# Optical pH Sensing in Milk: A Small Puzzle of Indicator Concentrations and the Best Detection Method

**Olga Voskoboynikova [1], Aleksey Sukhanov [1,\*] and Axel Duerkop [2]**

[1] School of Computer Science & Robotics of Tomsk Polytechnic University, Lenin Avenue 30, 634050 Tomsk, Russia; vostok@tpu.ru
[2] Institute of Analytical Chemistry, Chemo and Biosensors, University of Regensburg, Universitätsstraße 31, 93053 Regensburg, Germany; axel.duerkop@chemie.uni-regensburg.de
\* Correspondence: suhanov@tpu.ru

**Abstract:** Optical chemical sensors can yield distinctively different responses that are dependent on the method applied for readout and evaluation. We therefore present a comprehensive study on the pH determined non-continuously with optical sensors in real milk samples by either photometry or colorimetry (via the RGB-readout of digital images) compared to the pH values obtained electrochemically by potentiometry. Additionally, the photometric determination of pH was conducted with single-wavelength and a dual wavelength ratiometric evaluation of the absorbance. It was found that both the precision and accuracy of the pH determined by photometry benefit from lower concentrations of bromocresol purple, which served as the pH indicator inside the sensor membrane. A further improvement is obtained by the ratiometric evaluation of the photometric sensor response. The pH values obtained from the colorimetric evaluation, however, gain in precision and accuracy if a higher concentration of the indicator is immobilized inside the sensor membrane. This has a major impact on the future fabrication of optical pH sensor membranes because they can be better tuned to match to the most precise and accurate range of the planned detection method.

**Keywords:** pH sensor; optical; photometry; colorimetry; potentiometry; digital image

## 1. Introduction

Optical pH sensors have gained a key role in pH measurements. The basic principles of the creation of optical pH sensors and recent trends of readouts have been presented in comprehensive reviews [1,2]. Colorimetric pH sensors have become especially popular among optical pH sensors due to their low equipment demands. The signal of colorimetric sensors relies on the variation of color that is dependent on the pH of the analyzed sample. This enables simple readout schemes with printed tables for color comparison that can easily be operated also by non-laboratory users.

One of the trend directions for applying colorimetric pH sensors is in the freshness control of perishable foods. In food packaging, a freshness control of perishable foods is frequently not performed in a lab, instead an expiration date is usually stamped onto the packaging with additional recommendations for storage. This does not consider the effect of various conditions upon later transport and storage, which can significantly alter the shelf-life of perishable foods. The spoilage of perishable foods is accompanied by the formation of various chemical or biological species (e.g., biogenic amines or bacteria) or the change of certain characteristics of the foods (e.g., the color, odor and/or homogeneity), which can serve as freshness indicators that can be determined using the headspace above, in contact with, or inside perishable foods. Moreover, the intrinsic acid-base properties of the new compounds formed upon ageing will change the overall pH of the food sample, and their presence can hence be determined by means of a colorimetric pH sensor. Examples of applying colorimetric pH sensors for freshness control include seafood [3–8], meat products [9–11], dairy [12–17], and other perishable foods [18].

Milk acidity can serve as marker of its age, as can be seen from a sensor comprising a polypyrrole/silver nanocomposite chemically deposited on a polyester fiber [17]. The sensor had a reasonable response and reproducibility towards milk pH. Unfortunately, such fiber-optic sensors are not applicable for the evaluation of milk spoilage by untrained consumers outside the lab. Extracts from grape skin incorporated into a tara gum/cellulose nanocrystal matrix was recently used for visible pH-sensing and to monitor the milk spoiling process [13]. The proposed sensor films showed easily detectable color changes from red to green to monitor the freshness of packed food at acidic or alkaline pH, respectively. A new colorimetric method for the detection of milk spoilage using cysteine biofunctionalized silver nanoparticles was developed [15] where the aggregation of biofunctionalized nanoparticles increased with the increase of the lactic acid content of the sample, which leads to a color change from yellow to orange to red to purple. Further, colorimetric sensors based on an anthocyanin-agarose film were proposed as freshness indicators of full cream milk by means of monitoring the pH [16]. A new colorimetric method called red chromatic shift was applied to evaluate the color response of the proposed sensors towards pH. The activity of lactic acid bacteria leads to an increasing production of lactic acid and thus a decrease in the pH level. At normal levels of pH, the main protein in milk, casein, remains evenly dispersed. At lower pH levels below 4.6, the protein coagulates due to the acid generated from fermentation [19]. Thus, milk acidity can be used to monitor milk ageing.

We have previously demonstrated the maintenance of the acid-base properties of bromocresol purple (BCP) in a polymethacrylate matrix(PMM) and calculated the $pK_a$ value of the BCP in the PMM [20]. The response time of the PMM with immobilized BCP towards the changing pH of an aqueous solution was also determined. A polymethacrylate matrix was taken to tightly immobilize BCP and allow easy permeation of the analyte. There are good examples of applying PMM for the solid-phase quantification of various species [21–25]. The initial 6.5 pH value of spoiled milk [26] coincides with BCP's $pK_a$ value $6.5 \pm 0.3$ in the PMM. Therefore, PMM with immobilized BCP is suitable to monitor even small changes in the pH of milk. It is known that the visibility of a color transition depends on the concentration of an indicator [27]. Obviously, a specific amount of a sensitive reagent in a sensor membrane can also have an effect on its analytical performance.

In this article, various pH-sensitive sensor membranes based on bromocresol purple immobilized into a polymethacrylate matrix were prepared. Spectrophotometric one-wavelength and ratiometric dual-wavelength evaluations of the sensor response in milk were performed, and the resulting pH values were validated against those obtained electroanalytically using a pH electrode. Additionally, the colorimetric evaluation of digital images was compared to the potentiometric pHs. This yields new insights for choosing an appropriate amount of the pH indicator to achieve optimum accuracy and precision depending on the chosen detection method.

## 2. Materials and Methods

### 2.1. Apparatus and Chemicals

The absorption spectra and absorbances of sensor membranes and indicator stock solutions were acquired with a Shimadzu UV1800 spectrophotometer. Blanks of the related solvent or the untreated polymer membrane were subtracted. We used a I-160 ionometer (Izmeritelnaya Tekhnika NPO, Moscow, Russia) with a glass pH electrode to determine the reference pH values.

Lactic acid 85% (CAS No. 50-21-5) and bromocresol purple (CAS No. 115-40-2) were purchased from Chimmed, Russia. Those and all other chemicals were used as received and without further purification. The BCP stock solutions were created by dissolving accurately weighed portions of BCP (as to Table 1) in distilled water.

The polymethacrylate membrane was kindly provided by the group of Prof. Gavrilenko from Tomsk Polytechnic University and was synthesized according to a recent protocol [28]. Transparent 10 cm $\times$ 10 cm sheets with a thickness of $0.60 \pm 0.04$ mm were cut into working membrane pieces of 6.0 mm $\times$ 8.0 mm size and a mass of about 0.05 g.

**Table 1.** Parameters of sensors membranes dependent on immobilization conditions.

| Designation | (a [1] ± SD), mg/g | Immobilization Conditions | | | | Sensor Membrane | | | |
|---|---|---|---|---|---|---|---|---|---|
| | | $C_0$ [2], g/L | t, min | V, mL | m, g (Number of PMM Sheets) | $A_{415}$ [3] | RSD [4], % | $A_{600}$ [3] | RSD [4], % |
| SM1 | 1.21 ± 0.07 | 1.0 | 1.5 | 25 | 1.1316 (25 pcs) | 1.31 | 10 | 1.29 | 20 |
| SM2 | 0.82 ± 0.05 | 1.0 | 0.75 | 25 | 1.1287 (25 pcs) | 0.95 | 7 | 0.94 | 10 |
| SM3 | 0.268 ± 0.006 | 0.1 | 2.5 | 25 | 1.1356 (25 pcs) | 0.30 | 9 | 0.26 | 16 |

[1] amount of BCP immobilized under the appropriate conditions; [2] initial concentration of BCP in a solution for immobilization; [3] average absorption of sensor membranes at the 415 nm and 600 nm, respectively; [4] relative standard deviation in 25 membranes.

### 2.2. Preparation of the Sensor Membrane

The preparation of the sensor membrane followed a recent protocol [20]. First, pre-weighed PMM work sheets (25 pcs) were immersed into 25 mL of a stock solution of BCP with a definite concentration and shaken for a set time. Immobilization conditions are presented in Table 1. The amount of BCP immobilized into PMM was calculated according to

$$a = \frac{(C_0 - C) \times V}{m} \tag{1}$$

where $C_0$ and $C$ are concentrations of BCP in the solution before and after contact with PMM, respectively; $V$ is the volume of the solution used for immobilization, $m$ is the mass of the sheets (made of PMM) for immobilization. Prepared membranes were stored in a desiccator under dark conditions after the removal of excess BCP solution by dabbing the sensor membrane with filter paper.

### 2.3. Preparation of Milk Samples

Packed pasteurized cow milk (Derevenskoye molochko, Tomsk, Russia) was bought from a local supermarket. The method of the milk pasteurization was heating it to 90 °C for 20 s, which is termed as very high temperature (VHT) pasteurization according to the literature [29]. Milk samples with different pH were prepared with the addition of lactic acid dropwise to imitate a spoilage process [16]. All experiments using milk were done on the date of its manufacture as indicted on the package, at room temperature (24 ± 2 °C).

### 2.4. Procedure for Determination of pH in Milk Samples

Sensor membranes were immersed in 15 mL of milk samples with adjusted pH value and shaken for 5 min (according to preliminary investigations [20], the response time of similar membranes does not exceed 4 min). The related sensor membranes were then photographed, and the absorption spectra were recorded. Images were captured using a smartphone (Xiaomi Redmi 9) using the standard camera mode. The images were captured at a distance of 10 cm under constant vertical illumination by the built-in flash lamp. The pH of the milk samples was then immediately measured by means of a pH meter.

The algorithm for processing the images of the sensor membranes to extract the colorimetric parameters was created using the software "Vision builder for Automated Inspection (version 3.6.1, 2015)" by National Instruments. The total color difference (TCD) of the sensor membranes was calculated in the RGB color space according to the typical equation

$$TCD = ((R_0 - R_i)^2 + (G_0 - G_i)^2 + (B_0 - B_i)^2)^{1/2} \tag{2}$$

where the index "0" denotes the color parameter set of the sensor membrane after contact with the milk sample when the milk is fresh; index "*i*" denotes the color parameter set for the sensor membrane after contact with the milk sample with the adjusted pH. As shown in earlier work [20], such a sensor membrane is stable against repetitive cycling of pH between 1.5 and 11.3 in a flow-through cell for over 30 min. The sensors are not intended for prolonged and continuous contact with food.

## 3. Results

### 3.1. Preparation of Sensor Membranes

The amount of BCP in the PMM membrane depends on the immobilization conditions. A higher concentration of BCP in the solution for membrane doping leads to a higher concentration of BCP inside the sensor membrane (see Table 1). More BCP is also adsorbed by the sensor membrane when the immobilization time is increased. Compared with the stirring times usually required for the preparation of sensor cocktails for knife coating or electrospinning (several hours [30]), the current immobilization time is very short, being only a few minutes. This is very suitable for a potential mass production of sensors. According to their BCP content, the obtained sensor membranes were divided into three types. The characterization of the sensor membranes (as to photometric measurements) prepared under different immobilization conditions are presented in Table 1.

### 3.2. Spectrophotometric Response of Sensor Membranes towards Milk of Various pH

The response of a sensor membrane towards milk samples with various pH values was investigated by photometry. For this purpose, a sensor membrane was immersed into milk samples of adjusted pH and gently shaken. After 5 min, the sensor membrane was dabbed with filter paper, and the absorption spectrum was acquired in transmission mode. The related pH-dependent absorption spectra of the sensor membranes of the SM2 type after contact with the milk samples are presented in Figure 1. It is obvious that a decrease of pH (as it occurs upon the spoiling of the milk) is accompanied by an increase in the absorbance at 415 nm whereas a decrease in the absorbance is found at 600 nm. This is accompanied by a visual color transition from green to yellow. An isosbestic point is found at 485 nm. While the absorption spectra of BCP in a solution show maxima at 430 nm and 590 nm, those shift to 415 nm and 600 nm, respectively, after BCP immobilization in PMM. Please note that the additional reflected light (observed color) is determined by the molar absorbances at both maxima, which also experience ongoing change from the solution into the polymer [20]. This translates into the formation of a green mixing color on the sensor membranes at around pH 7. The absorption spectra of the other type of sensor membranes (SM1 and SM3) are similar with respect to the positions of the absorption maxima and the pH-dependent response. This suggests that the absorbance at these wavelengths or their ratio ($A_{415}/A_{600}$) may serve as analytical parameters for the solid-phase spectrophotometric determination for the pH of pasteurized milk. The color transition can also be used to visually follow the spoiling of milk.

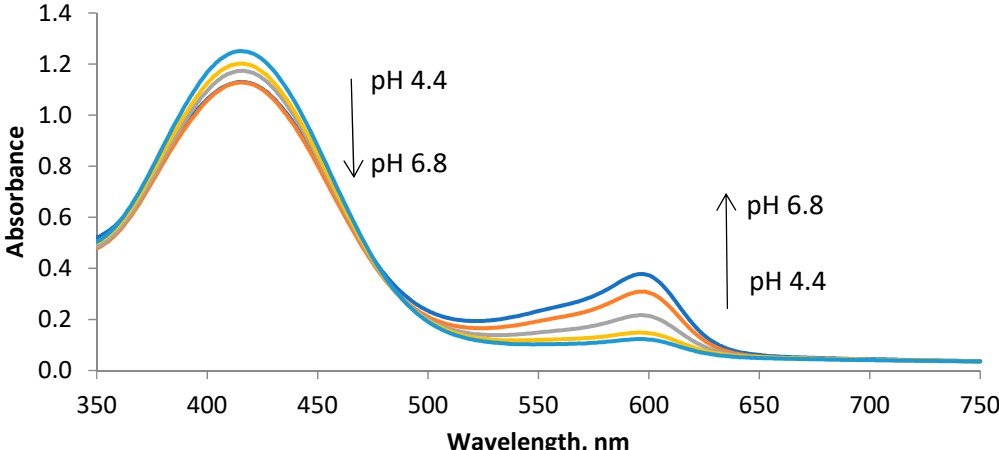

**Figure 1.** Absorption spectra of SM2-type sensors membranes after contact with milk samples at different pH.

### 3.3. Effect of the Content of BCP on Photometric Sensor Response

The changes of the photometric response of the sensor membranes with the changing pH of the milk samples should obey the Lambert–Beer law, both at 415 nm and at 600 nm. Therefore, the responses of the three different membrane types towards the milk pH were plotted at both wavelengths, and linear calibration plots were constructed (see Figure S1). A dependence of the analytical signal can be characterized by the linear equation

$$A = k \times pH + C \qquad (3)$$

where A is the analytical signal (i.e., the absorbance at the respective wavelength), pH is the pH value of the milk sample, and C is a constant. The related values of k and C for each analytical signal and the corresponding correlation coefficient (r) for each type of sensor membrane are presented in Table 2.

**Table 2.** Linear response of the analytical signal of the sensor membranes towards milk pH at various detection wavelengths and related correlation coefficients of the linear fit.

| Type of Sensor Membrane | k | C | r |
|---|---|---|---|
| Analytical signal: absorbance at 415 nm | | | |
| SM1 | −0.1225 | 2.3498 | 0.914 |
| SM2 | −0.039 | 1.3751 | 0.927 |
| SM3 | −0.0109 | 0.4889 | 0.858 |
| Analytical signal: absorbance at 600 nm | | | |
| SM1 | 0.2099 | −0.7004 | 0.983 |
| SM2 | 0.0982 | −0.3503 | 0.976 |
| SM3 | 0.0303 | −0.0757 | 0.992 |
| Analytical signal: $A_{415}/A_{600}$ | | | |
| SM1 | −1.7544 | 13.671 | 0.959 |
| SM2 | −2.9821 | 23.48 | 0.994 |
| SM3 | −1.3798 | 12.546 | 0.998 |

As the shortwave absorbance of BCP decreases with pH, a negative slope is found. The use of the absorbance at 415 nm as the analytical signal for the determination of milk pH is undesirable, as low correlation coefficients are found (see Table 2). Moreover, the overall change in the signal is less pronounced at this wavelength (see Figure 1). The absorbance at 600 nm and the ratio of the absorbance at 415 nm to the absorbance at 600 nm ($A_{415}/A_{600}$), however, can be used as an analytical signal for the solid phase spectrophotometric determination of milk pH. As for the correlation coefficients, $A_{600}$ shows much better results for all three compositions of the sensor membrane. The best correlation is found for the ratiometric response of $A_{415}/A_{600}$ that is dependent on milk pH. Here, SM2 and SM3 are the preferred membrane compositions.

### 3.4. Validation of Photometric Sensor Response

The pH values obtained from the photometric response of the sensor membranes was then validated against the pH acquired with a commercial pH-meter. As both the absorbance at 600 nm and $A_{415}/A_{600}$ had proved to be suitable for the determination of milk pH, both were used to judge the accuracy of the pH values determined with the sensor membranes. The obtained results are presented in Table 3. The maximum value of the relative error for every type of sensor membrane is indicated in bold. It is clearly obvious that membrane SM1 performs less well than membranes SM2 and SM3, if the related relative errors and the deviations between the average pH-vales as to potentiometry and photometry, respectively, are compared. Here, deviations up to 0.34 pH units are found. These deviations are much smaller for membrane SM2, where the deviation does not exceed 0.23 pH units (if potentiometry and $A_{600}$ are compared) or 0.17 pH units,

respectively ($A_{415}/A_{600}$ vs. potentiometry). Membrane SM3 offers the best accuracy, where deviations of less than 0.15 and 0.07 pH units ($A_{415}/A_{600}$ vs. potentiometry) are found. The latter deviation is already within the error range of the electrochemical pH determination, which is why the pH values determined with both methods can be regarded as equal. The average standard deviation of the photometric determination of pH over the whole range of pH 4.5–7.0 can be as little as 0.2 pH units (for membrane SM2), which is remarkably low for a colorimetric sensor. As a result, both accuracy and reproducibility of the sensor membrane SM2 and SM3 can be regarded very good.

**Table 3.** Validation of the pH obtained in milk by spectrophotometry with the pH from potentiometry ($n = 3$).

| Type of SM | pH Determined by Potentiometry | pH Determined by Spectrophotometry | | | |
|---|---|---|---|---|---|
| | | $A_{600}$ | $\delta^{600}$ | $A_{415}/A_{600}$ | $\delta^{415/600}$ |
| SM1 | $6.75 \pm 0.03$ | $6.73 \pm 0.73$ | 0.3 | $6.57 \pm 0.14$ | 2.6 |
| | $6.28 \pm 0.01$ | $6.17 \pm 0.07$ | 1.7 | $6.24 \pm 0.12$ | 0.6 |
| | $5.83 \pm 0.05$ | $6.07 \pm 0.31$ | **4.1** | $6.18 \pm 0.29$ | **6** |
| | $5.35 \pm 0.06$ | $5.27 \pm 0.37$ | 1.5 | $5.31 \pm 0.30$ | 0.8 |
| | $4.89 \pm 0.03$ | $4.86 \pm 0.31$ | 0.6 | $4.65 \pm 0.54$ | 4.9 |
| SM2 | $6.91 \pm 0.12$ | $7.16 \pm 0.34$ | 3.5 | $6.82 \pm 0.11$ | 1.3 |
| | $6.43 \pm 0.20$ | $6.26 \pm 0.39$ | 2.7 | $6.44 \pm 0.22$ | 0.1 |
| | $5.71 \pm 0.07$ | $5.59 \pm 0.15$ | 2.2 | $5.88 \pm 0.17$ | **2.9** |
| | $5.16 \pm 0.17$ | $5.00 \pm 0.06$ | 3.1 | $5.08 \pm 0.22$ | 1.6 |
| | $4.55 \pm 0.16$ | $4.78 \pm 0.10$ | **4.9** | $4.52 \pm 0.08$ | 0.7 |
| SM3 | $7.01 \pm 0.08$ | $7.04 \pm 0.23$ | 0.4 | $6.96 \pm 0.14$ | 0.8 |
| | $6.69 \pm 0.03$ | $6.74 \pm 0.14$ | 0.8 | $6.72 \pm 0.41$ | 0.5 |
| | $6.07 \pm 0.07$ | $5.99 \pm 0.59$ | 1.4 | $6.14 \pm 0.24$ | **1.1** |
| | $5.37 \pm 0.03$ | $5.22 \pm 0.16$ | **2.8** | $5.31 \pm 0.33$ | 1 |
| | $4.78 \pm 0.06$ | $4.91 \pm 0.23$ | 2.7 | $4.79 \pm 0.16$ | 0.2 |

### 3.5. Colorimetric Analysis of RGB Readout

The absorbance changes at both 415 nm and 600 nm of the sensor membranes after contact with the milk samples visually translate into an observable color transition from green to yellow (see Figure 2b). This color transition can be analyzed by taking images with a digital camera and the subsequent evaluation of the data contained in the three color channels. There are several recent examples using the RGB readout of digital camera images for the quantification of color transitions [30,31]. In our evaluation scheme, the total color difference (TCD) derived from the individual color differences in all three channels was used as the analytical signal (see Section 2.4) in the colorimetric analysis of pH (see Figure 2a). This was done for the three membrane types, which is why the effect of the BCP content on the quality of the RGB readout of sensor membranes could be derived. Equations describing the TCD response from the colorimetric analysis that is dependent on the pH of milk and the corresponding correlation coefficients are presented in Table 4. For the SM3 sensor membrane, a meaningful determination of pH using a linear equation is not possible although a difference between pH 7.0 and pH values less than 6.5 can visually be observed (Figure 2b). The fact that SM3 delivers a reliable result using spectrophotometry (as to Table 3) in contrast to the photographic readout is due to the differences in the light detection of both methods. Spectrophotometry detects the negative logarithm of the ratio of the intensity of the transmitted light emanating from the sample with respect to the intensity of the light irradiated into the sample. Both intensities are detected at a wavelength where the maximum change of light intensity is expected. This makes photometry amenable to detect even small changes of the absorbance of a sample i.e., of the pH. The photographic evaluation, however, uses the differences of the intensities of the reflected light of three wavelength ranges (i.e., the RGB colors), each of them integrated over a broad wavelength range. Further, the photometric signal is acquired by the absorption of light within the

entire volume of the sensor membrane. The colorimetric analysis of the image only uses the light reflected from the surface of the sensor, i.e., from a much lower volume of the sensor membrane. Finally, the sensitivity of the photomultiplier in the photometer is higher than the one of a CCD chip in a commercial digital camera. All of this contributes to a lower overall working range of SM3 with photographic readout.

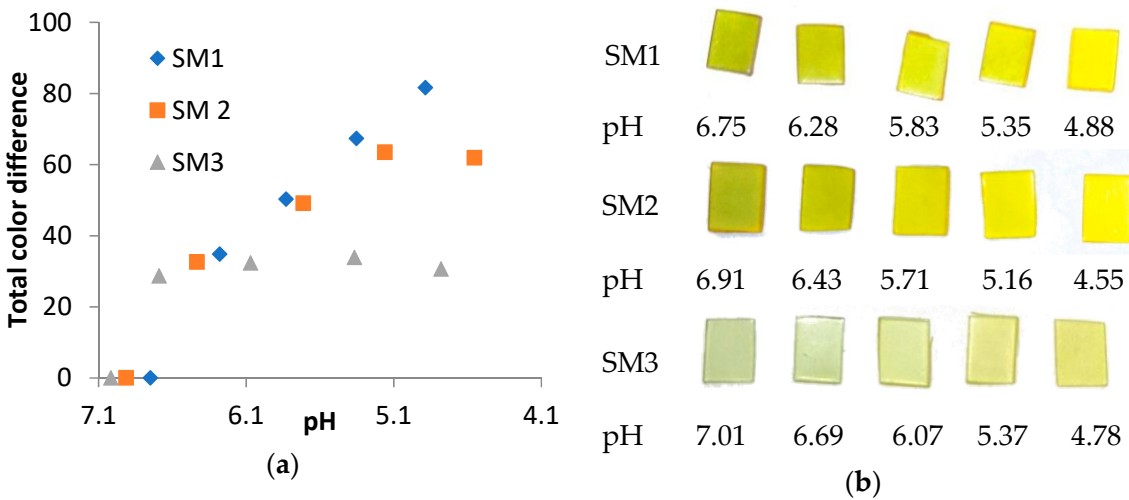

**Figure 2.** Colorimetric response of sensor membranes: (**a**) relationship of the total color difference (TCD) versus pH of milk samples; (**b**) corresponding images of the sensor membranes.

**Table 4.** Equations describing the dependence of the TCD on pH.

| Type of SM | Range of pH | Equation | *r* |
|---|---|---|---|
| SM1 | 6.75–4.88 | $TCD = -42.059\,pH + 291.6$<br>$TCD = -13.275\,pH^2 + 112.38\,pH - 151.85$ | 0.981<br>0.996 |
| SM2 | 6.91–5.16<br>6.91–5.16 | $TCD = -34.077\,pH + 242.62$<br>$TCD = -15.667\,pH^2 + 155.03\,pH - 321.02$ | 0.967<br>0.990 |
| SM3 | Not available | Linear equation not applicable | N/A |

Higher contents of BCP in a sensor membrane are accompanied by a wider pH range that correlates linearly to the TCD data. This is also obvious from the best correlation coefficient found for the linear evaluation of the TCD for membrane SM1. Hence the widest measurement range for the colorimetric evaluation via TCD is found for membrane SM1. The pH-range of SM2 is smaller. Using a 2nd order polynomial fit further improves the results of the colorimetric pH evaluation via TCD for SM1 and SM2 to yield very good correlation coefficients that are better than 0.990.

### 3.6. Validation of Colorimetricmetric Sensor Response

We then compared the pH values determined by colorimetric analysis via the TCD with the potentiometric data (see Table 5). A very good correlation was found specifically for membrane SM1 at all pH values except for the highest one (pH 6.75 vs. pH 6.94). Upon comparing the average deviation of the pH of membrane SM1 determined with either potentiometry or one of the two optical evaluation methods, it becomes clear that the determined colorimetrically determined pHs match the potentiometry data better than the photometry determined pH values (compare data of SM1 in Tables 3 and 5). The standard deviations of the pH determined via colorimetry and photometry, however, are very similar and are typically in a range of 0.1–0.3 pH units. The colorimetrically determined pH for membrane SM2 matches less well with the potentiometry data than that of membrane SM1.

Moreover, the pH-range for the colorimetric evaluation of SM2 is smaller (pH 5.2–7.0) than that of the same membrane evaluated with photometry (pH 4.5–7.0, see Table 3).

**Table 5.** Validation of the pH obtained in milk by colorimetry with the pH from potentiometry ($n = 3$).

| Type of SM | Potentiometric | Colorimetric | $\delta$,% |
|:---:|:---:|:---:|:---:|
| | $6.75 \pm 0.03$ | $6.94 \pm 0.13$ | 2.9 |
| | $6.28 \pm 0.01$ | $6.12 \pm 0.25$ | 2.6 |
| SM1 | $5.83 \pm 0.05$ | $5.70 \pm 0.44$ | 2.3 |
| | $5.35 \pm 0.06$ | $5.35 \pm 0.21$ | 0.06 |
| | $4.89 \pm 0.03$ | $4.99 \pm 0.13$ | 2.2 |
| | $6.91 \pm 0.12$ | $7.13 \pm 0.30$ | 3.1 |
| | $6.43 \pm 0.20$ | $6.16 \pm 0.22$ | 4.2 |
| SM2 | $5.71 \pm 0.07$ | $5.68 \pm 0.10$ | 0.7 |
| | $5.16 \pm 0.17$ | $5.25 \pm 0.11$ | 1.8 |

## 4. Conclusions

We demonstrated that the precision and accuracy of the pH determined with an optical sensor based on PMM with immobilized BCP in real milk samples is dependent on the readout method and the concentration of the pH indicator. Two quantification methods of pH were used: (i) spectrophotometry (single wavelength and double wavelength ratiometric) with a standard device and (ii) colorimetric readout based on the evaluation of the total color difference of the sensor images obtained by means of a smartphone camera. It was found that sensor membranes with high BCP content are more preferable for colorimetric pH determination, while for spectrophotometry, pH values gain in precision and accuracy when using lower concentrations of BCP inside the sensor membrane. The validation of the pH obtained by the optical readout of the sensor membrane against the electrochemically determined pH values with a glass electrode showed very similar values. In the case of the ratiometric evaluation of the photometric sensor response, the differences between electrochemical and optical readout were as low as 0.07 pH units, which is equal to the average standard deviation of both methods. Additionally, these sensor membranes can be used for the visual distinction between unspoiled and spoiled milk (pH of less than 6.0). This shows that the pH readings from optical pH sensors in real samples can reach the accuracy and precision of potentiometry as a reference method provided that the concentration of the indicator is tuned and adapted with respect to the optical readout method. A future application of these sensors could be monitoring water deacidification during the preparation of drinking water.

**Supplementary Materials:** The following are available online at https://www.mdpi.com/article/10.3390/chemosensors9070177/s1. Figure S1. Dependence of an analytical signal on the pH of the milk samples: (a) absorbance at 415 nm; (b) absorbance at 600 nm; (c) ratio $A_{415}/A_{600}$, $n = 3$, error bars represent standard deviation.

**Author Contributions:** Conceptualization, O.V. and A.S.; investigation, O.V. and A.S.; methodology, A.S. and A.D.; validation, O.V. and A.D.; writing—original draft, A.S. and A.D.; writing—review and editing, A.S. and A.D. All authors will be informed about each step of manuscript processing, including submission, revision, revision reminders, etc., via emails from our system or assigned Assistant Editor. All authors have read and agreed to the published version of the manuscript.

**Funding:** This research was supported by TPU development program.

**Conflicts of Interest:** The authors declare that they have no known competing financial interests or personal relationships that could appear to influence the work reported in this paper.

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
