# Peer review of "Optical pH Sensing in Milk: A Small Puzzle of Indicator Concentrations and the Best Detection Method"

_chemosensors, doi:10.3390/chemosensors9070177_

Round 1

Reviewer 1 Report

Measurement of pH in milk is important for detecting impurities, spoilage and signs of mastitis infection. While many factors can affect the composition of milk, pH measurements can help producers understand what may be causing some of these changes. There are still few studies in this area, so this study is not only interesting but also of practical value. Therefore, I would recommend its publication in Chemosensors as it fits well within the scope of the journal, after minor revisions. My comment at this respect is the following:

  1. Line 139, index ”1” should be index”i”.
  2. The pH range of SM2 can reach 4.5 which is not reasonable, because it is clear that total color difference reaches saturation at pH 5.1 in Figure 2a. Unless the authors can repeat the experiment three times to prove that the pH range of SM2 can be from 6.91 to 4.55, it is suggested to delete the description and equation of pH measurement range to 4.5 in Table 4.
  3. In Figure 2b, please adjust the brightness of the three pictures to the same level, so that the readers can easily make a comparison by the color change. In addition, the pH value of the color change (from left to right) should be marked on the picture.
  4. Although SM3 can measure different pH values in Table 3, the results of pH measurement in real milk samples are not satisfactory (Figure 2a). The authors should discuss the possible reasons for such a result in the article.
  5. Besides the measurement of pH value in the milk, what are the future applications?

Author Response

  1. We agree with the remark. The authors thank the referee for carefully reading the manuscript. The corresponding corrections were included in the manuscript.
  2. As pointed out by the referee, the response of the SM2 sensor membrane does not change at a pH less than 5.1. Nevertheless, a 2nd polynomial fit improves the correlation coefficient that can increase accuracy of the colorimetric analysis. The corresponding corrections were included on page 7 in the text and in table 4 in the manuscript.

  3. The corresponding corrections of brightness were included in figure 2b and the related pH vales were added below each sensor image.
  4. We added at the end of section 3.5: “The fact that SM3 delivers a reliable result using spectrophotometry (as to Table 3) in contrast to the photographic readout is due to the differences in light detection of both methods. Spectrophotometry detects the negative logarithm of the ratio of the intensity of the transmitted light emanating from the sample with respect to the intensity of the light irradiated into the sample. Both intensities are detected at a wavelength where the maximum change of light intensity is expected. This makes photometry amenable to detect even small changes of the absorbance of a sample .i.e. of pH. The photographic evaluation, however, uses the differences of the intensities of the reflected light of three wavelength ranges (i.e. the RGB colors), each of them integrated over a broad wavelength range. Further, the photometric signal is acquired by absorption of light within the whole volume of the sensor membrane. The colorimetric analysis of the image only uses light reflected from the surface of the sensor, i.e. from a much lower volume of the sensor membrane. Finally, the sensitivity of the photomultiplier in the photometer is higher than the one of a CCD chip in a commercial digital camera. All this contributes to a lower overall working range of SM3 with photographic readout.”
  5. A future application could be monitoring of water deacidification in the preparations of drinking water. This was added at the end of the conclusion.

Reviewer 2 Report

Optical methods for determining the quality of perishable foods, which make it possible not to violate the integrity of the packaging, are currently an actively developing area, therefore the topic of the work is relevant. The authors developed pH-sensitive sensor membranes based on bromocresol purple immobilized in a polymethacrylate matrix and demonstrated their applicability for spectrophotometric and colorimetric measurements.

The manuscript is clearly structured and I can recommend it for publication, but after fixing some of the issues:

  1. The work lacks information on the assessment of photostability and resistance to dye leaching. Evaluation of these parameters is a must during the development of optical sensors. It is especially important for sensors that aimed to contact with food. This is especially important for sensors aimed at contact with food, because this is the type of sensor being developed, judging by the introduction (lines 42-45).
  2. The pKa value of the indicator in water solution is 6.3 [https://pubchem.ncbi.nlm.nih.gov/compound/8273], and above pH 6.8 it has a violet color. According to the preliminary study of the authors (ref 20), immobilization practically does not shift pKa=6.5 ± 0.3. Figure 2 shows photo of membranes series with pH from 4.5 to 7.0, however, visually, none of the samples even comes close to violet. In the aforementioned ref. 20 such a transition is observed, and at pH 7 the color change is visible. What is the reason for the discrepancy between the results of these works?
  3. The direction of the abscissa axis (pH) in Fig. 2 and S1 is different, which is confusing. It is worth signing the pH under the samples (Fig 2. b), and indicate in the text the direction of the pH change which causes the "observable color transition from green to yellow" (lines 224-225).
  4. The best sample in spectrophotometric determination was SM3, and the worst was SM1. Colorimetric determination demonstrates completely opposite results. The authors simply state the fact, but do not provide any explanation for the observed phenomenon.
  5. In Fig. 1, for clarity, it is worth indicating the direction of change in the spectrum components, as is done, for example, in work [10.3103/S0027131421010090, Fig. 2].
  6. The pH and Absorbance axes in supplementary information and in Fig. 1 should be formatted to display the same number of significant digits (up to tenths).
  7. Formulas on lines 110 and 137 should be given on separate lines, like formula (1) on line 177. Either the formula for calculating the total color difference, or its explanation contains an error. The formula uses the index “i”, the text refers to “1”.
  8. Ambiguous sentence: “A polymethacrylate matrix was taken to immobilize BCP due to its capabilities to tightly bind the reagent and extract the analyte to be determined.” (lines 71-73). It may appear to the reader that the sensory membrane matrix is extracting something from the milk during the analysis. This is not true?
  9. There are a number of typos, e.g. line 74 “quantification OF various species”, line 83 “pH values”, line 93 “untreated”. The entire manuscript should be checked.
  10. Strange phrase “dipping the sensor membrane onto a paper towel.” (line 114). “dabbed with filter paper” (line 159) is much better.

Author Response

  1. We agree with the reviewer. Yet, photo degradation and dye leaching was not found during previous investigations [ref 20] and in the present investigation. We therefore added to section 2.4.: “As shown in earlier work [20], such a sensor membrane is stable over 30 minutes against repetitive cycling of pH between 1.5 and 11.3  in flow-through cell.”
  2. We agree with the reviewer that a violet color is observed in aqueous solution of BCP at pH 7. At the same time, absorption spectra of BCP in a solution show maxima at 430 nm and 590 nm. After immobilization BCP in PMM however, both absorption maxima are shifted to 415 nm and 600 nm, respectively (see figures 2a and b in ref 20). Hence, the difference of the two maxima is increased to 185 nm. This change of both positions of the two maxima translates into a green color of the sensor membranes with respect to the color of BCP observed in solution. Please note that the reflected light (observed color) is additionally determined by the molar absorbances at both maxima which also change on going from solution into the polymer ((see figures 2c and d in ref 20). Both effects contribute to the formation of a green mixing color of the sensor membranes at around pH 7. The blue-violet color of the sensor membranes is indeed observed at pH values higher than 7.5. Moreover, this dichroism of bromocresol purple has been proposed to make it a difficult indicator in colorimetric pH determination of a transparent solution [Bishop, Indicators, V.1].

    We added as explanation to section 3.2:

    “While the absorption spectra of BCP in a solution show maxima at 430 nm and 590 nm, those shift to 415 nm and 600 nm, respectively, after immobilization BCP in PMM. Please note that the reflected light (observed color) is additionally determined by the molar absorbances at both maxima which also change on going from solution into the polymer [20]. This translates into the formation of a green mixing color of the sensor membranes at around pH 7.”
  3. The direction of the abscissa axis in Fig.2 corresponds to the change in pH during milk aging. In Fig. S1 were made appropriate corrections. The pH values are now indicated below the images.
  4. Please see our comment to the related comment #4 of reviewer 1. We added in section 3.5 as explanation:

    “The fact that SM3 delivers a reliable result using spectrophotometry (as to Table 3) in contrast to the photographic readout is due to the differences in light detection of both methods. Spectrophotometry detects the negative logarithm of the ratio of the intensity of the transmitted light emanating from the sample with respect to the intensity of the light irradiated into the sample. Both intensities are detected at a wavelength where the maximum change of light intensity is expected. This makes photometry amenable to detect even small changes of the absorbance of a sample,.i.e. of pH. The photographic evaluation, however, uses the differences of the intensities of the reflected light of three wavelength ranges (i.e. the RGB colors), each of them integrated over a broad wavelength range. Further, the photometric signal is acquired by absorption of light within the whole volume of the sensor membrane. The colorimetric analysis of the image only uses light reflected from the surface of the sensor, i.e. from a much lower volume of the sensor membrane. Finally, the sensitivity of the photomultiplier in the photometer is higher than the one of a CCD chip in a commercial digital camera. All this contributes to a lower overall working range of SM3 with photographic readout.”
  5. We agree with the remark. The corresponding corrections were included in the manuscript.

  6. We agree with the remark. The corresponding corrections were included in the manuscript.

  7. We agree with the remark. The corresponding corrections were included in the manuscript.

  8. We agree that this sentence was potentially misleading and rephrased as follows: “A polymethacrylate matrix was taken to tightly immobilize BCP and allow easy permeation of the analyte.”
  9. We thank the referee for carefully reading the manuscript. The corresponding corrections were included in the manuscript.
  10. We thank for this suggestion. The corresponding corrections were included in the manuscript.

Round 2

Reviewer 2 Report

The authors significantly improved the manuscript and corrected almost all issues. However, the main issue regarding sensor stability test has not been fixed. A reference to the results of the preliminary study (ref. 20) is not applicable in the case of the subject of the present work (milk freshness control).

The introduction says (lines 38-44): "The spoilage of perishable foods is accompanied by the formation of various chemical or biological species (e.g. biogenic amines or bacteria), or the change of certain parameters of foods (e.g. the color, odor and/or homogeneity), which can serve as freshness indicators and can be determined in the headspace over or in contact with or inside perishable foods. Moreover, the intrinsic acid-base properties of the new compounds formed upon ageing will change the overall pH of the food sample, and their presence can hence be determined by means of a colorimetric pH sensor."

Further (lines 78 onwards): "In this article, various pH-sensitive sensor membranes based on bromocresol purple immobilized into a polymethacrylate matrix were prepared."

After reading the introduction, the reviewer got the impression that the sensor created by the authors is suitable for incorporation into product packaging for direct control of freshness. Homemade (farm) milk can be stored in the refrigerator for up to 4 days, pasteurized for 7-10 days, sometimes even 14 days. Accordingly, stability tests should last not 30 minutes (as in the preliminary work, ref. 20), but at least several days, and preferably weeks. In a similar work [10.3103/S0027131421010090], no photodegradation was observed for 30-60 minutes experiment; nevertheless, within several days of continuous measurements, it becomes noticeable.

It is highly doubtful that the sensor using a simple adsorption principle (current work) does not show any dye leaching. If the created sensor is not suitable for prolonged contact with food, or is not intended for this, then this should be clearly stated in the article. However, in the latter case, the question arises about the expediency of the work done. How do the authors propose to check the freshness of milk using the created sensor? For example, one will open the package and dip the sensor inside? In this case, it is easier to use a traditional potentiometric sensor. However, if the properties of the sensor meet the stated purpose of measuring acidity without opening the package and without contamination, then it is worth providing evidence.

Author Response

It seems that the reviewer obviously got a wrong impression about the major objectives of our paper. Although we cite papers in the introduction that also describe sensors for continuous measurements of pH in milk, we never state or propose our sensor to be intended for continuous use or use in packages. It is obvious from the paper headline and the abstract and the last paragraph of the introduction that the intention of this paper is to compare accuracy and precision of various detection methods of sensors that were used for determination of the pH of milk. We further state that optimized sensor compositions were achieved for various detection methods. Finally, it is obvious from section 2.4 that our sensor principle aims at non-continuous sampling (dipping of the sensor into milk sample for 5 min) with subsequent determination of the pH. In so far, we do not give the impression that a sensor for continuous or in-package use is described.

Yet, in order to consider the concern of the reviewer, we added a note to the abstract, that our sensors are intended for non-continuous measurement.

We agree with the opinion of the reviewer that at a certain point of time there will be leaching of the indicator off the sensor membrane. However, within the stated response time of 5 min and with the known pH stability from earlier work [20] and using noncontinuous sampling (please see comment above) dye leaching will be hardly relevant in this work.
Yet, in order to consider the concern of the reviewer, we added a note to the abstract, that our sensors are intended for non-continuous measurement.

We indeed consider detecting pH in milk packages by colorimetry in future work. However, this will require a completely different sensor regime. This will require determination of the pH (from the sensor without removing it from the package) by colorimetry in remission mode. Hence, the sensor surface facing milk must be permeable for protons whereas the backside sensor surface must be attached milk-impermeable but to a transparent polymer support to be read from outside the package. Therefore, a sensor for continuous monitoring of milk freshness in a package, as pointed out by the reviewer, is a separate scientific work, which involves not only studying the impact of the sensor contact on product safety (Leaching, photobleaching, loss of polymer into the milk package, etc.), but also developing an appropriate way to integrate the sensor into the packaging material. Yet, it should be pointed out again, that this was not the aim of this study. We did a study on various detection methods and sensor properties, as indicated above and also in the abstract and in the conclusion.

We also do not state the purpose of measuring milk acidity without opening the package and without contamination. We do not even refer to in-package sensing. In so far, we also do not see, why we would need to start a discussion on if it would be better to measure milk pH potentiometrically or colorimetrically after opening of the package. To consider the concerns of the reviewer we added “The sensors are not intended for prolonged continuous contact with food.” in section 2.4.

Although we cannot fully meet the expectations of the reviewer, we hope that we could work out that the issue of leaching is not relevant for the short contact time of our sensors with milk and that we clarified the sensor properties with respect to its non-continuous operation. Fabrication of a continuous in-package sensor was and is not stated as the aim of this work but rather a study to choose and validate an appropriate detection method and sensor composition for the detection of the pH of milk.